# The Effect of Loneliness on Nomophobia: A Moderated Mediation Model

**DOI:** 10.3390/bs13070595

**Published:** 2023-07-16

**Authors:** Shupeng Heng, Qiwei Gao, Minghui Wang

**Affiliations:** 1Faculty of Education, Henan Normal University, Xinxiang 453007, China; 15071366473@163.com (S.H.); gaoqiwei029@163.com (Q.G.); 2Faculty of Education, Henan University, Kaifeng 475001, China

**Keywords:** loneliness, nomophobia, smartphone attachment, attachment anxiety

## Abstract

A large number of studies have shown that loneliness is a positive predictor of nomophobia (no mobile phobia), but little research has examined the mechanism of this association. Drawing on attachment theory, the present study constructs a moderated mediation model to examine whether smartphone attachment mediates the relation between loneliness and nomophobia and whether this mediating process is moderated by attachment anxiety. A total of 598 college students in China were recruited to complete a questionnaire to measure loneliness, smartphone attachment, attachment anxiety, and nomophobia. A moderated mediation analysis was conducted. Our findings revealed that after controlling for gender, loneliness is positively associated with nomophobia. Meanwhile, smartphone attachment plays a mediating role in the relationship between loneliness and nomophobia. Furthermore, attachment anxiety moderates the mediated path through smartphone attachment, such that an indirect effect is much stronger for individuals with higher attachment anxiety relative to those with low attachment anxiety. The present study provides new insight into the complex processes in the association between loneliness and nomophobia, and the results have important theoretical and practical implications.

## 1. Introduction

With the development of network and communication technology, smartphones have become an indispensable part of modern life. The powerful functions of smartphones meet people’s various needs, change people’s behaviors and habits [1], and even extend individuals’ self [2]. However, there is also a rise in excessive dependence on smartphones [3], resulting in nomophobia, a state in which individuals feel anxious and restless when they are unable to use or access smartphones [1,4]. Indeed, nomophobia is becoming increasingly prevalent [4,5,6], and there is evidence to suggest strong links between nomophobia and psychological problems [6]. Given the recent and alarming rise in nomophobia, this study sets out to explore its influencing factors and mechanisms. 

Loneliness is an emotional experience when an individual’s interpersonal relationships fail to meet his or her expectations. Loneliness is often caused by a lack of social interaction, social belonging, or intimate emotional attachment [7]. Individuals with a strong sense of loneliness lack close social interactions and extensive social support in real life, despite having a desire to be connected with others [8]. The accessibility and versatility of smartphones may not only help individuals to access online entertainment, but they may also help individuals to establish virtual social relationships. Therefore, individuals who feel lonely may see smartphones as a primary means of achieving a sense of connectedness and belonging. However, this may also be a double-edged sword in that such individuals may become excessively dependent on smartphones [9,10]. Indeed, one study has found that limited face-to-face interactions with others may lead lonely individuals to be particularly interested in what others are doing [11]. When such individuals are separated from their smartphones and have no means of keeping track of others online, they may experience a heightened sense of anxiety. Although a large number of studies have shown that loneliness is a positive predictor of nomophobia [6,12,13,14], it remains unclear just how or when loneliness affects nomophobia. This study seeks to address this gap by exploring the relationship between loneliness and nomophobia, including their influencing factors (moderating effect) and internal mechanisms (mediating effect).

### 1.1. The Mediating Role of Smartphone Attachment

Although several empirical studies have revealed the relationship between loneliness and nomophobia [6,12,14], it remains to be seen exactly how loneliness triggers an individual’s nomophobia. We contend that smartphone attachment plays a mediating role between loneliness and nomophobia, and we now outline the concept of smartphone attachment, a relatively new phenomenon.

Attachment refers to an emotional bond between infants and their caregivers, so once separated from their caregivers, infants would feel anxiety and discomfort [15]. When the caregiver is not around, children may form attachments to other people or things (such as plush toys) to compensate for the lack of attachment to their primary caregiver because these substitutes enable children to obtain similar feelings they would derive from their caregiver [16]. Similar to children, adults may also form attachments to fictional characters, pets, and objects that comfort them when needed [16].

As smartphones become increasingly powerful and their widespread use continues, people are increasingly inseparable from them and form attachments to their devices [17]. Drawing on attachment theory, Konok, Gigler, Bereczky, and Miklósi proposed the concept of smartphone attachment, which frames the smartphone as more than just a tool for communication; it may potentially become an attachment object that brings people a sense of security [18]. Indeed, many individuals seek to ensure that their smartphones are always on hand and go to great lengths to avoid the anxiety caused by being separated from their smartphones. Konok et al. [18] found that the dependence of individuals on smartphones has less to do with the device’s capacity to enable communication with others; rather, it is more about individuals perceiving the smartphone as an external attachment object. As with the attachment of infants to their caregivers, once an individual forms an emotional attachment to a smartphone, they will have a psychological “connection” with their smartphone, and the smartphone will become an attachment object that can bring security to him or her. The upshot of this is that individuals may experience a high degree of anxiety if they become separated from their smartphones [17,19,20].

Compensatory attachment strategies theory posits that when important social relationships (main attachment objects) are weakened or become temporarily unavailable or lost, individuals will form attachments to alternative objects. Therefore, when individuals perceive that the main attachment object is unreliable, they will trigger a compensatory attachment to the object [15,21]. Studies have shown that when individuals are in uncertain social relationships, their attachment to objects will increase, the pressure of separation from mobile phones will increase, and they have a strong motivation to contact mobile phones [22]. Socially alienated individuals tend to project human qualities onto non-human entities [23] and are more likely to use objects such as computers, cars [24,25], and even pets to meet their social needs [26]. As a tool to maintain relationships and social connections, smartphones are reliable, controllable, and ready to use as needed [16]. Hence, they are more likely to serve as compensatory attachment objects, replacing human social connections and offering a sense of security to individuals [16]. We contend that highly lonely individuals are more likely to use smartphones to meet their social needs and boost their emotions, which becomes a precursor to the formation of attachments to their smartphones. Therefore, we propose the following hypotheses. 

**Hypothesis 1.** *Smartphone attachment plays a mediating role in the relationship between loneliness and nomophobia*.

### 1.2. The Moderating Effect of Anxiety Attachment

Although loneliness will have a significant impact on nomophobia directly or indirectly, there may be differences in how this impact is realized. We set out to investigate whether the relationship between loneliness and nomophobia is moderated by other factors in a bid to determine the conditions in which loneliness plays a role in this relationship. In particular, we examine whether the direct and indirect effects of loneliness on nomophobia vary depending on the degree to which the individual exhibits attachment anxiety.

As previously discussed, attachment involves individuals developing a special emotional connection with specific objects or people [27]. Individuals with avoidant attachment have difficulties in establishing intimate relationships and often seek to maintain independence and emotional distance from others [28]. In contrast, individuals with attachment anxiety tend to have minimal self-worth, fear rejection, and abandonment, are hypersensitive to the response of others, and have a higher need for intimacy [29]. People with a high level of attachment avoidance tend to be more self-contained and self-sufficient, whereas those with high levels of anxiety attachment are more dependent [18,30]. One study revealed that those with high avoidance attachment are more averse to having pets compared to those with high anxiety attachment [31]. Equally, people with high attachment anxiety tend to see smartphones as objects of attachment [18], are more likely to become addicted to their smartphones [32,33], and have a greater risk of nomophobia when they are separated from their phones [34]. 

Since people with high anxiety attachment tend to be more dependent and crave the approval and attention of others, they are more likely to adopt compensatory attachment strategies to gain a sense of security [29]. The ability of smartphones in enabling individuals to communicate with others and, thus strengthen one’s relationships, is especially important for individuals with high attachment anxiety. Having formed an attachment to their smartphones, such individuals show a higher degree of anxiety when they are separated from their phones [18]. Thus, we propose the following research hypothesis.

**Hypothesis 2.** 
*Attachment anxiety is a moderator of the relationship between smartphone attachment and nomophobia. Speci*
*fically, for individuals with high anxiety attachment, smartphone attachment has a significant positive predictive effect on nomophobia. For individuals with low anxiety attachment, smartphone attachment has no significant predictive effect on nomophobia.*


In summary, this study develops a moderated mediation model (see Figure 1) in exploring the effects of loneliness, attachment anxiety, and smartphone attachment on nomophobia, and how these factors interact with each other. 

## 2. Methods

### 2.1. Participants

A total of 598 college students (19.0% male, 81.0% female) were recruited by facilitating sampling from four universities in China. Their ages ranged from 17 to 23 years old (*M* = 19.42, *SD* = 1.75). Participants completed a survey that collected information about their demographic variables in addition to measuring their levels of loneliness, attachment anxiety, smartphone attachment, and nomophobia.

### 2.2. Measures

*Loneliness*. Loneliness was measured by the Chinese version of the UCLA loneliness self-rating scale [35]. The scale includes 20 items, and a sample item is “Do you often feel the lack of partners?” Participants rated each item on a four-point scale (1 = *I’ve never felt that way*, 4 = *I often feel that way*), and Cronbach’s α was 0.90 in this study. 

*Adult Attachment*. Adult attachment was measured with a translated version of the Adult Attachment Scale (AAS) [36]. The scale consists of three subscales: intimacy (six items), dependence (six items), and anxiety (six items). Our study used the anxiety subscale to measure the degree to which an individual fears being abandoned or unliked. A sample item of this subscale is “I want to be close to people, but I worry about being hurt”. Participants rated each item on a five-point scale (1 = *complete non-compliance* and 5 = *full compliance*), and Cronbach’s alpha was 0.75 in the current study.

*Mobile Attachment*. Mobile attachment was measured using the ten-item Mobile Attachment Scale (MAS) developed by Konok et al. [18]. The scale consists of two subscales: having one’s phone in close proximity and feeling stressed when separated from one’s phone. A sample item from this scale is “I regularly check my phone even if it does not ring”. Participants rated each item on a five-point scale (1 = *complete non-compliance* and 5 = *full compliance*). Cronbach’s alpha in the current study was 0.85.

*Nomophobia*. Nomophobia was measured with a translated version of the Nomophobia Scale for Chinese (NMP-C) [37]. The scale includes four subscales: fear of being unable to obtain information, fear of losing convenience, fear of losing contact, and fear of losing connection to the Internet. The scale consists of 16 items, such as “I feel uncomfortable if I can’t continuously access information through my phone”. Participants rated each item on a seven-point scale (1 = *Complete non-compliance* and 7 = *full compliance*). Cronbach’s alpha in the current study was 0.95.

### 2.3. Data Analysis

In this study, spss 25.0 was used for data analysis. We used a correlation matrix to describe the current status of college students’ loneliness, smartphone attachment, attachment anxiety, and nomophobia. Moreover, we used the Process Macro to examine the interaction and mediation effects. 

### 2.4. Ethics

This study was approved by the institutional review board of the Institute of Education, Henan Normal University, China. Informed consent was obtained from all participants prior to assessment.

## 3. Results

### 3.1. Descriptive Analysis

The means, standard deviations, and correlational analyses for all variables are presented in Table 1. Loneliness was positively correlated with smartphone attachment and nomophobia (*r* = 0.18, *p* < 0.01; *r* = 0.12, *p* < 0.05). In addition, smartphone attachment was positively correlated with nomophobia (*r* = 0.57, *p* < 0.01). Finally, attachment anxiety was positively associated with nomophobia (*r* = 0.08, *p* < 0.05).

### 3.2. Testing for Moderated Mediation

According to the results of the correlation analysis, the relationships between loneliness, attachment anxiety, smartphone attachment, and nomophobia meet the requirements of the moderated mediation model test. Studies have found significant gender differences in nomophobia [6,12]. Therefore, gender will affect the research results. But gender is not a variable to be examined in this study, so this study used gender as a control variable in the analysis of the results, and all predictors were mean centering to minimize multicollinearity [38]. As all predictors had variance inflation factors below two, multicollinearity did not exist in the study. The moderated mediating effects test was then completed in the SPSS macro prepared by Hayes [39]. More precisely, we conducted percentile bootstrapping as well as bias-corrected percentile bootstrapping with 5000 resamples to construct 95% confidence intervals for the indirect effects.

According to Muller, Judd, and Yzerbyt [40], it is necessary to test the parameters of the three regression equations to test the moderated mediation models. (1) The relationship between loneliness and nomophobia (Model 1); (2) the relationship between loneliness and smartphone attachment (Model 2); and (3) the relationship between smartphone attachment and nomophobia, as well as the residual effect of loneliness on nomophobia (Model 3). If one or both of the following conditions are met, the moderated mediation is established: (a) the path from loneliness to smartphone attachment is moderated by attachment anxiety and (b) the path from smartphone attachment to nomophobia is moderated by attachment anxiety. The specific results of the three models are shown in Table 2.

As Table 2 illustrates, Model 1 showed that both loneliness and attachment anxiety significantly predicted nomophobia (*β* = 0.11, *p* = 0.007 < 0.01; *β* = 0.03, *p* = 0.02 < 0.05), but the interaction effect was not significant (*β* = −0.13, *p* = 0.07 > 0.05). Model 2 indicated that the main effect of loneliness and attachment anxiety on smartphone attachment was significant (*β* = 0.08, *p* = 0.02 < 0.05; *β* = 0.32, *p* = 0.006 < 0.01), but the interaction effect was not significant (*β* = −0.02, *p* = 0.08 > 0.05). In Model 3, attachment anxiety and smartphone attachment both had a significant effect on nomophobia (*β* = 0.15, *β* = 0.60, *p* = 0.005, 0.007 < 0.01), and the interaction effect of smartphone attachment and attachment anxiety on nomophobia was also significant (*β* = 0.07, *p* = 0.02 < 0.05). The results of our regression analysis show that a moderated mediating effect model can explain the relationship between the variables, and smartphone attachment plays a partial mediating role in this model, with a mediating effect value of 0.13. At the same time, attachment anxiety plays a moderating role in the relationship between smartphone attachment and nomophobia. 

To more clearly demonstrate how attachment anxiety moderates the relations between mobile phone attachment and nomophobia, this study divides attachment anxiety into high and low groups (one *SD* above the mean and one *SD* below the mean, respectively; see Figure 2). Simple slope tests showed that, for individuals with low attachment anxiety, smartphone attachment did not significantly predict nomophobia (*β* simple = 0.11, *t* = 0.98, *p* > 0.05); however, for individuals with high attachment anxiety, smartphone attachment significantly predicted nomophobia (*β* simple = 0.38, *t* = 3.31, *p* < 0.01).

## 4. Discussion

This study explored the relationship between loneliness and nomophobia and its underlying mechanisms. The results show that a moderated mediation model can be used to explain the relationships between loneliness, attachment anxiety, smartphone attachment, and nomophobia. Not only does loneliness directly affect nomophobia but it also has an indirect effect on nomophobia through smartphone attachment, which is particularly significant for individuals with high anxiety attachment.

### 4.1. Loneliness and Nomophobia

We also find that loneliness positively predicts nomophobia. The higher the level of loneliness, the higher the level of nomophobia, which is consistent with the results of previous studies [6,12,14]. This suggests that individuals who are very lonely are more likely to use smartphones to increase their contact with others and achieve a sense of belonging through online interactions [9,10]. Such individuals are at greater risk of developing excessive dependence on smartphones. Because their sense of belonging and connectedness is so reliant on online interactions [41], they typically experience greater anxiety if they are separated from their smartphones.

### 4.2. The Mediating Role of Smartphone Attachment

In addition to finding that loneliness and smartphone attachment positively predict nomophobia, our study also reveals the mediating effect of smartphone attachment on the relationship between loneliness and nomophobia, supporting our first hypothesis. Our finding is consistent with previous research that indicates that individuals with high levels of loneliness are more inclined to emotionally invest and form attachments to inanimate objects [24,25]. The psychological connection individuals form with their smartphones leads to anxiety should they become separated from their devices [17,19,20].

According to the theory of compensatory attachment strategies, the compensatory attachment to objects is triggered when individuals perceive the primary attachment object (e.g., the lack of access to important social relations) is unreliable [21]. For individuals with high loneliness, their sense of belonging and social needs are difficult to meet due to the lack of necessary social interaction and social support, so they are more likely to meet their social needs through non-human entities [24,25,26]. As a tool to maintain relationships and social contacts, smartphones can replace people’s social contacts and provide a sense of security, which makes them more likely to become compensatory attachment objects [16]. Therefore, a smartphone is not only a tool for communication but also an attachment object that can bring a sense of security. That explains why individuals always seek to stay close to their phones to avoid the anxiety caused by separation from phones [18].

### 4.3. The Moderating Effect of Attachment Anxiety

Although smartphone attachment can significantly affect nomophobia, we found that levels of attachment anxiety have a moderating effect on the relationship between smartphone attachment and nomophobia. For individuals with high levels of attachment anxiety, smartphone attachment significantly predicts nomophobia, whereas smartphone attachment has no significant predictive effect for those with low levels of attachment anxiety. As such, our second hypothesis was supported. 

Our results are in line with previous studies that individuals with high levels of attachment anxiety are more emotionally dependent on interpersonal relations [18], people, objects, or events [30]. This also shows that individuals with high anxiety attachment tend to use compensatory attachment strategies to gain a sense of security [29], and the attachment function of smartphones (relationship promotion function and compensatory attachment to objects) is more important for them. Therefore, once they form attachments to smartphones, they will feel more fearful and uncomfortable when they are separated from smartphones (attachment objects) [18]. This study not only examines smartphone attachment’s impact on nomophobia but also further explains when the mediating effect is stronger by using a moderated mediation model. It also effectively integrates individuals’ experiences of emotional attachment with types of attachment. In short, this study deepens and expands the research on the relationship between smartphone use and individuals’ psychosocial adjustment.

### 4.4. Research Significance and Future Recommendations

This study has two important theoretical implications. First, it integrates attachment theory [15] and compensatory attachment strategy theory [21] to explain how loneliness affects nomophobia through smartphone attachment. Second, this study shows the applicability of smartphone attachment to wider theories of attachment. It illustrates when and how the mediation model becomes stronger or weaker and deepens and expands the mediation model. This integration improves the explanatory power of the model, which is exactly where the value of a moderated mediation model is different from a simple mediation or moderated model. At the same time, the findings of this study have a certain practical significance in the prevention and intervention of nomophobia among people who are lonely. That is, those who are socially isolated can reduce both their dependence on smartphones and their levels of anxiety caused by separation from smartphones by seeking to increase their face-to-face interactions with other people so as to meet their need for connections and belonging.

The limitations of this study should be addressed. First, the sample type is relatively single, and external validity is low in this study, so it is necessary to expand the type of sampling to achieve external validity. Second, this study is a cross-sectional study, so it is impossible to infer the causal relationship between variables. In future research, longitudinal or experimental research can be used to verify the results of this study. Third, future researchers can also explore the mediating role of other important variables (e.g., self-expansion) between loneliness and nomophobia. Fourth, this study only focused on the moderating role of attachment anxiety in the relationship between smartphone attachment and nomophobia, but future studies can explore the moderating role of other attachment types, such as avoidant attachment.

## 5. Conclusions

This study represents an important step in unpacking how loneliness predicts nomophobia. Our findings suggest that the positive impact of loneliness on nomophobia can be partially explained by smartphone attachment. Moreover, attachment anxiety moderates this indirect link in the second stage of the mediation process. Specifically, for high attachment anxiety individuals, the effect of smartphone attachment on nomophobia is particularly significant. By shedding light on the complex interplay among loneliness, attachment anxiety, smartphone attachment, and nomophobia, this study provides valuable empirical evidence that “how” and “under what conditions” loneliness increases nomophobia.

## Figures and Tables

**Figure 1 behavsci-13-00595-f001:**
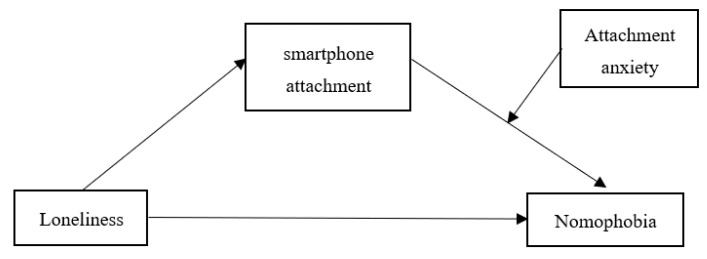
A model of the predictors of nomophobia.

**Figure 2 behavsci-13-00595-f002:**
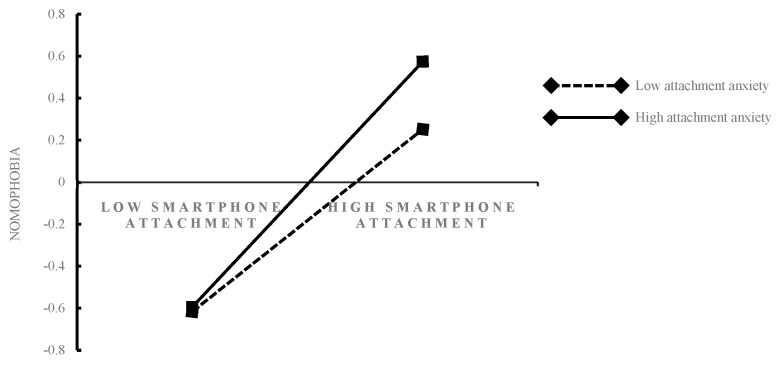
Attachment anxiety as a moderator of the relationship between smartphone attachment and nomophobia. The moderating effect is graphed for two levels of attachment anxiety: 1 standard deviation above the mean and 1 standard deviation below the mean.

**Table 1 behavsci-13-00595-t001:** Means, standard deviations, and correlations for variables (*n* = 598).

	*M*	*SD*	1	2	3	4
1 Loneliness	42.36	9.18	−			
2 Smartphone Attachment	34.27	7.16	0.18 **	−		
3 Attachment Anxiety	2.96	0.94	0.35 **	0.35 ***	−	
4 Nomophobia	66.5	19.3	0.12 *	0.57 **	0.08 *	−

**Note: *** *p* <0.05, ** *p* < 0.01, *** *p* < 0.001same below.

**Table 2 behavsci-13-00595-t002:** Testing the moderated mediation effects of loneliness on nomophobia.

	Model 1 (Nomophobia)		Model 2 (Smartphone Attachment)		Model 3 (Nomophobia)	
	*β*	*t*	*β*	*t*	*β*	*t*
Gender	0.16	3.99 **	0.17	4.40 **	0.05	1.49
Loneliness	0.11	2.59 **	0.08	1.86 *	0.06	1.75
Attachment Anxiety	0.03	1.70 *	0.32	7.79 **	0.15	4.11 **
Loneliness×Attachment Anxiety	−0.13	0.90	−0.02	−0.62	0.03	0.78
Smartphone Attachment					0.60	16.50 **
Smartphone Attachment×Attachment Anxiety					0.07	1.97 *
*R^2^*	0.04		0.15		0.37	
*F*	6.32 **		26.22 **		58.26 **	

Note: * *p* < 0.05. ** *p* < 0.01.

## Data Availability

The data are available from the corresponding author upon reasonable request.

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
