# Peer review of "The Effect of Loneliness on Nomophobia: A Moderated Mediation Model"

_behavsci, 2023, doi:10.3390/bs13070595_

Round 1
Reviewer 1 Report
1. Expand the type of sampling to give certainty of external validity.
2. Expand the sociodemographic data: types of university study.
3. How the participants were recruited, the period of time, the ethical considerations according to the APA code of ethics.
4. Time and daily use of the mobile (social uses (sociodigital networks, calls) and individual uses (games, apps, YouTube, other))
5. Loneliness is a construct defined by various authors. It is convenient to adopt the theory of the loneliness scale used to expand that social interactions can be complemented with sociodigital interactions, or even suggest an extension based on the natural use of the mobile.
Author Response
Response to Reviewer 1Comments
(Manuscript ID:behavsci-2478486)
Thanks to the valuable comments of the reviewers of the Behavioral Sciences. In the following, according to the comments and suggestions of reviewers, explain the changes one by one. The main revisions are marked in blue in the text.
Point 1: Expand the type of sampling to give certainty of external validity.
Response 1:Thank you very much for your valuable comment. The participants are college students in this study, so the sample type is relatively single and external validity is low. It is be necessary to expand the type of sampling to give certainty of external validity.
Point 2: Expand the sociodemographic data: types of university study.
Response 2: Thank you very much for your valuable comment. The participants were recruited from four universities in China, the sociodemographic data is relatively limited. According to your suggestion,we will expand the sociodemographic data,not just in university.
Point 3:How the participants were recruited, the period of time, the ethical considerations according to the APA code of ethics.
Response 3: The participants were recruited by facilitating sampling from four universities in China. The study was approved by the institutional review board of the Institute of Education, Henan Normal University, China. Informed consent was obtained from all participants prior to assessment.
Point 4: Time and daily use of the mobile (social uses (sociodigital networks, calls) and individual uses (games, apps, YouTube, other))
Response 4: I strongly agree with your viewpoint and thank you very much for your valuable comment. The daily use of mobile phones is very extensive, including social uses (sociodigital networks, calls) ,individual uses (games, apps, YouTube, other) and so on.
Point 5: Loneliness is a construct defined by various authors. It is convenient to adopt the theory of the loneliness scale used to expand that social interactions can be complemented with sociodigital interactions, or even suggest an extension based on the natural use of the mobile.
Response 5: I strongly agree with your viewpoint and thank you very much for your valuable comment. As you say,it is convenient to adopt the theory of the loneliness scale to expand that social interactions.
Reviewer 2 Report
The authors presented an interesting piece of work on smartphone attachment, loneliness and attachment. I think the article is overall clear and well written, and the topic studies is of interest for the journal. Data collection and Analysis seems to have been conducted correctly. I have some minor comments on the structure of the article, and in particular on the clarity of the reporting, which I have detailed below:
- Figure 1 is missing a caption.
- To make it easier to be found in the text, I would suggest adding (H1) and (H2) where your hypothesis are described on page 3.
- In the data analysis section, please also add the operating system on which SPSS was used for reproducibility.
- In section 3 Results, please report the full p-value, not just < x, as this allows for the evaluation of the effect size. The same applies to the p-values of the models (Line 226-229).
- On line 199, the authors should consider adding an explanation of why Gender is used a control variable.
- On line 295 it is claimed that attachment anxiety is a risk factor for nomophobia, however I don't believe the current data can show a casual relation, as the authors also claim in their limitation section. I believe the authors should carefully check their text in order to make sure that the conclusions thoroughly supported by the results
Overall I believe the article is of good quality.
I believe the Quality of English is good and minor grammar edits are required.
Author Response
Response to Reviewer 2Comments
(Manuscript ID:behavsci-2478486)
Thanks to the valuable comments of the reviewers of the Behavioral Sciences. In the following, according to the comments and suggestions of reviewers, explain the changes one by one. The main revisions are marked in blue in the text.
The authors presented an interesting piece of work on smartphone attachment, loneliness and attachment. I think the article is overall clear and well written, and the topic studies is of interest for the journal. Data collection and Analysis seems to have been conducted correctly. I have some minor comments on the structure of the article, and in particular on the clarity of the reporting, which I have detailed below:
Point 1:Figure 1 is missing a caption.
Response 1:Thank you very much for your valuable comment. According to your suggestion,the author has added a caption for Figure 1.
Point 2:To make it easier to be found in the text, I would suggest adding (H1) and (H2) where your hypothesis are described on page 3.
Response 2:Thank you very much for your valuable comment. According to your suggestion,the author has added H1and H2where hypothesis are described on page 3.
Point 3:In the data analysis section, please also add the operating system on which SPSS was used for reproducibility.
Response 3:Thank you very much for your valuable comment. According to your suggestion,the author has added the operating system.
Point 4:In section 3 Results, please report the full p-value, not just < x, as this allows for the evaluation of the effect size. The same applies to the p-values of the models (Line 226-229).
Response 4:Thank you very much for your valuable comment. According to your suggestion,the author has reported the full p-value.
Point 5:On line 199, the authors should consider adding an explanation of why Gender is used a control variable.
Response 5:Studies have found significant gender differences in nomophobia(Buctot, Kim ,& Kim, 2020; Burhanettin et al., 2018; Gezgin & Çakir, 2016; Gezgin, Cakir ,& Yildirim, 2018; Secur Envoy, 2012; Tavolacci et al., 2015; Yildirim et al., 2016).Therefore, gender will affect the research results.But gender is not a variable to be examined in this study, so this study used gender as a control variable in the analysis of results.
Point 6: On line 295 it is claimed that attachment anxiety is a risk factor for nomophobia, however I don't believe the current data can show a casual relation, as the authors also claim in their limitation section. I believe the authors should carefully check their text in order to make sure that the conclusions thoroughly supported by the results
Response 6:Thank you very much for your valuable comment. As reviewers say, the current data can't show a casual relation. According to your suggestion, the authors carefully check the text and restated the content of this section.
Reviewer 3 Report
Thank you very much for the opportunity to read this engaging study. The authors have presented the rationale behind this study using established attachment and object relations theory. I'd like to seek some clarifications regarding the paper:
Abstract: I think a structured abstract is not required.
l. 26, it is unclear how individual "selves" are affected by smartphones. Kindly elaborate.
l. 65 - I believe these "substitutes" are also called transitional objects. Can I confirm that?
l. 109 "when exactly does loneliness play a role..." I think there may be a better way to phrase this, as we are not looking at the temporal relationship.
l. 120 I feel that describing people as "needy" is rather stigmatising. May I request for the authors to change this word to a more objective word?
l. 121 Please change "more adverse" to "more averse"
Perhaps hypotheses are appropriate, since the literature has provided a rationale for the study. What do you think?
The cronbach's alpha reported are from this study, or from past studies? Kindly clarify.
The range of the scale for mobile attachment, adult attachment, and nomophobia (compliance, non-compliance) needs to be rechecked.
l. 199 There was inadequate explanation on why gender was used as a control variable. Were there past literature to back up this decision?
l. 200 - May I know why mean centering was needed?
Some parts of ll203-214 should be in the methods section, as the statistical analysis section does not fully portray the type of analyses used. Was Process Macro used, and did you use bootstrapping, and how many iterations, if it was employed? Kindly clarify if that is the case.
l. 332, I don't think "when" is something the authors investigated, as the temporal relationship was not investigated. Kindly clarify.
Minor edits, especially spacing between words is not consistent.
Author Response
Response to Reviewer 3Comments
(Manuscript ID:behavsci-2478486)
Thanks to the valuable comments of the reviewers of the Behavioral Sciences. In the following, according to the comments and suggestions of reviewers, explain the changes one by one. The main revisions are marked in blue in the text.
Thank you very much for the opportunity to read this engaging study. The authors have presented the rationale behind this study using established attachment and object relations theory. I'd like to seek some clarifications regarding the paper:
Point 1:Abstract: I think a structured abstract is not required.
Response 1: Thank you very much for your valuable comment. According to your suggestion,
the authors have changed into unstructured abstract.
Point 2:l. 26, it is unclear how individual "selves" are affected by smartphones. Kindly elaborate.
Response 2:Thank you very much for your valuable comment. According to your suggestion,we modify it as“and even extend individuals’ self[2]”.
Point 3:l. 65 - I believe these "substitutes" are also called transitional objects. Can I confirm that?
Response 3: Yes, It can also be understood in this way.
Point 4:l. 109 "when exactly does loneliness play a role..." I think there may be a better way to phrase this, as we are not looking at the temporal relationship.
Response 4: Thank you very much for your valuable comment. As you say, there isn’t temporal relationship. According to your suggestion,we modify it as “in a bid to determine under what conditions does loneliness play a role in this relationship”.
Point 5:l. 120 I feel that describing people as "needy" is rather stigmatising. May I request for the authors to change this word to a more objective word?
Response 5:Thank you very much for your valuable comment. As reviewers say, "needy" is rather stigmatising. According to your suggestion,we modify it as “whereas those with high levels of anxiety attachment are more dependent”.
Point 6:l. 121 Please change "more adverse" to "more averse"
Response 6:Thank you very much for your valuable comment. According to your suggestion,we change "more adverse" to "more averse".
Point 7:Perhaps hypotheses are appropriate, since the literature has provided a rationale for the study. What do you think?
Response 7:I Agree with your viewpoint and thank you for your affirmation. As you say, the literature has provided a rationale for the study,so the hypotheses are appropriate.
Point 8:The cronbach's alpha reported are from this study, or from past studies? Kindly clarify.
Response 8:Thank you very much for your valuable comment.The cronbach's alpha reported are all from this study. According to your suggestion,we clarify them.
Point 9:The range of the scale for mobile attachment, adult attachment, and nomophobia (compliance, non-compliance) needs to be rechecked.
Response 9:Thank you very much for your valuable comment. According to your suggestion,we rechecked all the range of the scale.
Point 10:l. 199 There was inadequate explanation on why gender was used as a control variable. Were there past literature to back up this decision?
Response 10: Thank you very much for your valuable comment. According to your suggestion,the author has added an explanation of why gender is used as a control variable.
Studies have found significant gender differences in nomophobia(Buctot, Kim ,& Kim, 2020; Burhanettin et al., 2018; Gezgin & Çakir, 2016; Gezgin, Cakir ,& Yildirim, 2018; Secur Envoy, 2012; Tavolacci et al., 2015; Yildirim et al., 2016).Therefore, gender will affect the research results.But gender is not a variable to be examined in this study, so this study used gender as a control variable in the analysis of results.
Point 11:l. 200 - May I know why mean centering was needed?
Response 11: According to Frazier,Tix and Barron(2004), to reduce problems with multicollinearity among the continuous variable ,the independent variables were mean centered for the regressions.
Point 12:Some parts of ll203-214 should be in the methods section, as the statistical analysis section does not fully portray the type of analyses used. Was Process Macro used, and did you use bootstrapping, and how many iterations, if it was employed? Kindly clarify if that is the case.
Response 12:Thank you very much for your valuable comment. According to your suggestion,we portray the type of analyses used.
The moderated mediating effects test was then completed in the SPSS macro prepared by Hayes[39]. More precisely, we conducted percentile bootstrapping as well as bias-corrected percentile bootstrapping with 5000 resamples to construct 95% confidence intervals for the indirect effects.
Point 13:l. 332, I don't think "when" is something the authors investigated, as the temporal relationship was not investigated. Kindly clarify.
Response 13:Thank you very much for your valuable comment. As you say, there isn’t temporal relationship. According to your suggestion,we modify it as “it provides a more comprehensive understanding of “how” and “under what conditions” loneliness may increase nomophobia.”.